# The Effects of Reverse Nordic Exercise Training on Measures of Physical Fitness in Youth Karate Athletes

**DOI:** 10.3390/jfmk9040265

**Published:** 2024-12-10

**Authors:** Raja Bouguezzi, Senda Sammoud, Yassine Negra, Younés Hachana, Helmi Chaabene

**Affiliations:** 1Research Laboratory (LR23JS01) “Sport Performance, Health & Society”, Manouba 2010, Tunisia; rajabouguezzi@hotmail.com (R.B.); senda.sammoud@gmail.com (S.S.); yassinenegra@hotmail.fr (Y.N.); hachanayounes@gmail.com (Y.H.); 2Institut Supérieur de Sport et de l’Éducation Physique du Kef, Université de Jendouba, Le Kef 7100, Tunisia; 3Higher Institute of Sport and Physical Education of Ksar Saïd, University of “La Manouba”, Manouba 2037, Tunisia; 4Department of Sport Science, Chair for Health and Physical Activity, Otto-von-Guericke University Magdeburg, 39106 Magdeburg, Germany

**Keywords:** combat sport, athletic performance, eccentric training, physical conditioning, injury

## Abstract

**Background:** In karate, the ability to execute high-velocity movements, particularly kicks and punches, is heavily dependent on the strength and power of the lower limb muscles, especially the knee extensors. As such, this study aimed to evaluate the effects of an 8-week eccentric training program utilizing the reverse Nordic exercise (RNE) integrated into karate training compared with regular karate training only on measures of physical fitness in youth karate athletes. **Methods:** Twenty-seven youth karatekas were recruited and allocated to either RNE group (n = 13; age = 15.35 ± 1.66 years; 7 males and 6 females) or an active control group ([CG]; n = 14; 7 males and 7 females; age = 15.30 ± 1.06 years). To track the changes in measures of physical fitness before and after training, tests to assess linear sprint speed (i.e., 10 m), change of direction (CoD) speed (i.e., modified 505 CoD), vertical jumping (i.e., countermovement jump [CMJ] height) and horizontal jumping distance (i.e., standing long jump [SLJ]), and lower-limb asymmetry score (i.e., the difference between SLJ-dominant and non-dominant legs) were carried out. **Results:** The results indicated significant group-by-time interactions in all measures of physical fitness (effect size [ES] = 1.03 to 2.89). Post-hoc analyses revealed significant changes in the RNE group across all performance measures (effect size [ES] = 0.33 to 1.63). Additionally, the asymmetry score exhibited a moderate decrease from pre to posttest (∆46.96%, ES = 0.64). In contrast, no significant changes were observed in the CG across all fitness measures. Moreover, the individual response analysis indicated that more karatekas from the RNE group consistently achieved improvements beyond the smallest worthwhile change threshold across all fitness measures. **Conclusions:** In summary, RNE training is an effective approach to enhance various physical fitness measures besides lower-limb asymmetry scores in youth karatekas and is easy to incorporate into regular karate training. Practitioners are therefore encouraged to consistently integrate RNE training to enhance essential physical fitness components in young karatekas.

## 1. Introduction

Karate is a dynamic sport that requires a combination of strength, power, flexibility, agility, and endurance for optimal performance [1,2]. The ability to execute high-velocity movements, particularly kicks and punches, is heavily dependent on the strength and power of the lower limb muscles, especially the knee extensors [1,3]. Thus, incorporating strength training, especially targeting the knee extensors, is essential for performance enhancement in karate.

Eccentric training, characterized by the lengthening of muscles under tension, is a well-established method for improving muscle strength and physical fitness [4,5,6,7]. It has been shown to produce greater gains in muscle mass and strength compared with other forms of muscle action (e.g., concentric only, isometric only, or concentric–eccentric) [4,5,8,9]. This appears to be, in some part, due to higher anabolic signaling, as reflected by the greater satellite cell activation following eccentric compared with concentric muscle actions [10]. Additionally, eccentric training places high mechanical stress on muscles, promoting adaptations that enhance muscle tendon properties and boost overall physical fitness [4,11]. This makes it an effective means for athletes aiming to increase their general physical conditioning. Moreover, eccentric training has been recognized as an effective tool for reducing both the frequency and severity of sports-related injuries [12,13,14].

Previous eccentric training studies have primarily been conducted with young adults [4]. Douglas and Pearson [4] indicated that the range of age of participants included across their 40 considered studies was between 18 and 35 years, with the vast majority of studies (80%) including untrained participants and only 20% including either resistance-trained or moderately/highly trained participants. This highlights a substantial void in the literature in terms of the lack of studies involving youth athletes. A recent systematic review advocated for the inclusion of eccentric training into youth athletes’ training programs to improve a wide range of physical fitness measures such as muscle strength, jumping, linear sprint speed, and CoD (change of direction) speed [15]. The same authors indicated that Nordic hamstring exercises and flywheel inertial training are currently the most used eccentric resistance training methods in youth athletes [15], highlighting the limited choice of (accentuated) eccentric exercises available and the need for a more varied selection of effective exercises. Moreover, the use of a flywheel device per se cannot consistently guarantee the achievement of an eccentric overload as many factors come into play like the type of exercise, inertial load used, training experience, and braking techniques [16], substantiating the need for alternative, more accessible eccentric exercises.

Despite the proven efficacy of eccentric training in various sports [4,7,17], the application of specific eccentric exercises like the reverse Nordic exercise (RNE) remains underexplored [15]. The RNE is a specific, easy-to-implement eccentric exercise targeting the quadriceps muscles, which are vital for many lower-limb movements in karate [1,3]. To date, only one study has examined the effects of the RNE on the architectural adaptations of the rectus femoris in young adults aged 24 years [18]. The findings indicated that after 8 weeks, two sessions per week of RNE training generated a large increase in muscle fascicle length, muscle thickness, pennation angle, and cross-sectional area [18]. The structural adaptations indicated in the study of Alonso-Fernández and Fernández-Rodríguez [18] imply that the RNE has the potential to improve general physical fitness attributes, such as lower-limb muscle power and CoD speed, that are beneficial across many athletic disciplines, including karate. However, this is yet to be empirically proved.

In this context, incorporating eccentric exercises like the RNE into training routines can help karate athletes improve key physical fitness components. As such, this study aimed to evaluate the effects of an 8-week intervention utilizing the RNE compared with regular training only on measures of physical fitness in youth karate athletes. Building on findings from previous investigations [15,17], it was hypothesized that an 8-week intervention using the RNE would improve measures of physical fitness and reduce asymmetry scores in youth karate athletes.

## 2. Materials and Methods

### 2.1. Experimental Approach to the Problem

This study examined whether 8 weeks of a biweekly in-season RNE training program would enhance various measures of physical fitness in youth karate athletes relative to their peers who maintained their customary in-season training regimen. Two groups from two regional karate teams were recruited. Both groups underwent four karate training sessions per week. Two weeks before baseline testing, two familiarization sessions were performed to get participants acquainted with the applied tests. Several tests were used to track changes in physical fitness before and after the training program. These included tests for linear sprint speed (i.e., 10 m), CoD speed (i.e., modified 505 CoD), vertical jumping (i.e., countermovement jump [CMJ] height), horizontal jumping (i.e., standing long jump [SLJ], SLJ-dominant leg, and SLJ-non-dominant leg), and lower-limb asymmetry score. All tests were scheduled at least 48 h after the last training session at the same time of day (19:00–20:30).

### 2.2. Participants

With reference to a previous study [17], an a priori power analysis using G*Power software (version 3.1.9.7) was conducted, setting a type I error rate at 0.05 and aiming for 80% statistical power. The analysis indicated that overall, 14 participants would be sufficient to detect a significant effect, with an effect size (Cohen’s d) of 0.85 for the CMJ height. To account for potential participant attrition, 27 youth karatekas were recruited and allocated to either an RNE group (n = 13; 7 males and 6 females) or an active control group ([CG]; n = 14; 7 males and 7 females) (Table 1). All participants were experienced karatekas, with an average of 6.0 ± 1.2 years of systematic karate training. Furthermore, all participants were in good health and had been free of musculotendinous injuries for the six months preceding the study. The biological age of participants was estimated using the maturity offset method by applying the following prediction equations [19]:Males: maturity offset = −7.999994 + (0.0036124 × age × height)
Females: maturity offset = −7.709133 + (0.0042232 × age × height)

All experimental procedures and potential risks were thoroughly explained to both participants and their parents/legal guardians. Before the study began, written informed consent and assent from the parents and participants were obtained. The experimental procedure was approved by the local Institutional Review Committee of the “blinded for review” (LR23JS20) and conducted per the latest version of the Declaration of Helsinki.

### 2.3. Linear Sprint Speed Time

An electronic timing system assessed 10 m linear sprint performance (Wittygate, Microgate, SRL, Bolzano, Italy). Participants began in a standing split stance position, with their lead foot positioned 0.3 m behind the first infrared photoelectric gate, which was set 0.75 m above the ground to capture trunk movement and minimize false signals caused by limb motion. A total of two single-beam photoelectric gates were used. Participants were instructed not to rock or take false steps before starting the sprint. A recovery time of three minutes was allowed between trials, and the best performance out of two trials was used for further analysis. The between-trial intraclass correlation coefficient (ICC_3,1_) was 0.91.

### 2.4. The Modified 505 Change of Direction Speed

During the modified 505 CoD speed test, karatekas were instructed to perform a 5 m sprint from a starting line, place their preferred foot on the 5 m line, turn 180°, and sprint back 5 m through the start/finish line. Single-beam infrared photocell gates (Wittygate, Microgate, SRL, Bolzano, Italy) were positioned 0.75 m above the ground at the start line. A resting period of three minutes was provided between trials. The best performance out of two trials was used for further analysis. The between-trial ICC_3,1_ was 0.88.

### 2.5. Countermovement Jump Height

During the CMJ, participants started from a standing position and executed a fast downward movement by flexing the knees and hips before rapidly extending the legs and performing a maximal vertical jump. During the test, participants were instructed to maintain their arms akimbo. Vertical jump height was recorded using an optoelectric system (Optojump next, Microgate, SRL, Bolzano, Italy). A rest period of one minute was allowed between trials. The best out of two trials was retained for further analysis. The between-trial ICC_3,1_ was 0.94.

### 2.6. Standing Long Jump Distance

The starting position of the SLJ required participants to stand with their feet shoulder-width apart behind a starting line and their arms loosely hanging down. On the command “ready, set, go”, participants executed a countermovement with their legs and arms and jumped at maximal effort in the horizontal direction. Participants had to land with both feet at the same time and were not allowed to fall forward or backward. The horizontal distance between the starting line and the heel of the rear foot was recorded via a tape measure to the nearest 1 cm. A rest period of one minute was allowed between trials, and the best out of two trials was recorded for further analysis. The between-trial ICC_3,1_ was 0.92.

### 2.7. Asymmetry Score

To determine the asymmetry score, the SLJ-dominant leg and SLJ-non-dominant leg tests were conducted while participants adopted a standing position on the designated testing leg, with their hands on their hips and their toes behind the starting line. The dominant leg is identified as the preferred leg that the practitioner uses for executing kicks. Participants were then instructed to hop forward as far as possible and land on the same leg. Upon landing, participants were required to “hold and stick” their position for ~2 s. The mean inter-limb asymmetry was calculated as a percentage difference between limbs in the unilateral tests using the following equation: (100/[maximum value] × [minimum value] × −1 + 100) [20]. Two trials were conducted for each leg, and the best was retained for further analysis. The between-trial ICC_3,1_ results were 0.92 and 0.93 for the SLJ-dominant leg and SLJ-non-dominant leg, respectively.

### 2.8. Reverse Nordic Training Program

The training intervention consisted of a progressive 8-week eccentric strengthening program using the RNE (Table 2). The training intervention consisted of two sessions per week, performed right after the standard warm-up and replacing 10 to 20 min of low-intensity karate drills with the RNE on Tuesday and Thursday (Table 2). The training load was progressively increased from 2 to 4 sets per session and 6 to 10 repetitions per set throughout the intervention period. To perform the RNE, from a kneeling position with arms crossed over the chest, participants were instructed to engage their abdominals and glutes to stabilize their core and keep the torso upright. By maintaining the hips extended and the torso firm, participants slowly leaned backward by bending their knees (Figure 1). Participants were instructed to lower themselves as far as they could while maintaining control, then return to the starting position by straightening their knees. The intensity was progressively raised by increasing the range of motion of the backward lean over time. The CG followed their usual karate training routine. Therefore, the total training exposure was comparable between the two groups.

### 2.9. Statistical Analyses

Data are presented as means and standard deviations (SDs). The normality assumption was tested and confirmed using the Shapiro–Wilk test. To establish the effect of the interventions on the dependent variables, a 2 (group: RNE group and CG) × 2 (time: pre, post) ANOVA with repeated measures was determined for each parameter. When group × time interactions reached the level of significance (i.e., significant F value), group-specific post-hoc tests (i.e., paired *t*-tests) were used. The alpha level of significance was set at *p* < 0.05. To determine the magnitude of the training effect, effect sizes (ES) were determined by converting partial eta-squared to Cohen’s d. According to Hopkins and Marshall [21], ES values are classified as trivial (<0.2), small (0.2–0.6), moderate (0.6–1.2), large (1.2–2.0), very large (2.0–4.0), and extremely large (>4.0). The smallest worthwhile change (SWC) was calculated as 0.2 × SD pooled, where SD represents the pooled standard deviation of pre-training scores. Between-trial reliability was assessed using the ICC. All data analyses were performed using SPSS 25.0 (SPSS, Inc., Chicago, IL, USA).

## 3. Results

The anthropometric data for both groups are displayed in Table 1. All participants received the treatment as allocated. No training or test-related injuries were reported. All physical fitness measures at baseline and follow-up are presented in Table 3. At baseline, no significant between-group differences were observed with respect to anthropometric characteristics and maturity offset. The maturation level of all participants was postpubertal (Table 1). Similarly, no between-group differences were recorded at baseline for any measure of physical fitness (Table 3).

### 3.1. Linear Sprint Speed

Our findings indicated a significant main effect of time for the 10 m sprint (ES = 2.94 [large], *p* < 0.001). In addition, a significant group × time interaction was observed (ES = 2.89 [large], *p* < 0.001) (Table 3). Post-hoc analyses showed a large pre–post performance improvement in the RNE group (∆10.09%; *p* < 0.001; ES = 1.63). However, the CG did not show any significant change (∆-0.10%; *p* > 0.05; ES = 0.0 [trivial]). In terms of the individual responses, our statistical analysis indicated that 100% of the RNE group (n = 13) improved their 10 m sprint performance to a level that was greater than the SWC_0.2_ compared with only 53.84% in the CG (n = 7).

### 3.2. Change of Direction Speed

For the modified 505 CoD test, the results indicated a significant main effect of time (ES = 1.59 [large], *p* < 0.001). Similarly, the group-by-time interaction was significant (ES = 1.94 [large], *p* < 0.001) (Table 3). Post-hoc analyses demonstrated moderate pre–post 505 CoD speed performance decrements for the RNE group (∆-7.07%; *p* < 0.01; ES = 1.18). No significant pre–post performance changes were found for the CG (∆0.67%; *p* > 0.05; ES = −0.1 [trivial]). The individual response analysis indicated that 100% of the RNE group (n = 13) improved their 505 CoD speed performance to a level that exceeded the SWC_0.2_ compared with only 7.14% in the CG (n = 1).

### 3.3. Vertical Jumping

For CMJ height, a significant main effect of time was observed (ES = 1.96 [large], *p* < 0.001). Likewise, a significant group-by-time interaction was noted (ES = 1.43 [large], *p* < 0.01) (Table 3). Post-hoc analyses demonstrated a small pre–post CMJ height improvement for the RNE group (∆13.55%; *p* < 0.01; ES = 0.57). No significant pre–post changes were observed in the CG (∆2.20%; *p* > 0.05; ES = 0.09 [trivial]). In terms of individual responses, our statistical calculations showed that 76.92% of the RNE group (n = 10) improved their CMJ height to a level that exceeded the SWC_0.2_ compared with only 12.29% (n = 2) of the CG.

### 3.4. Horizontal Jumping

For the SLJ test, a significant main effect of time (ES = 1.66 [large], *p* < 0.01) and group-by-time interaction (ES = 1.32 [moderate], *p* < 0.05) were observed. Post-hoc analyses indicated a trivial pre–post improvement for the RNE group (∆4.80%; *p* < 0.01; ES = 0.33), with no significant difference detected in the CG (∆0.55%; *p* > 0.05; ES = 0.04 [trivial]). The individual response analysis revealed that 61.54% of the RNE group (n = 8) improved their SLJ test performance to a level that was greater than the SWC_0.2_. However, all the athletes in the CG failed to reach changes that went beyond the SWC_0.2_.

### 3.5. Asymmetry Score

Regarding the asymmetry scores, the statistical analysis revealed a large effect of time (ES = 2.94; *p* < 0.01) with a moderate group-by-time interaction (ES = 1.03, *p* < 0.05). Post-hoc analysis showed a moderate pre–post decrement in the RNE group (∆46.93%; ES = 0.64; *p* < 0.05), while the CG failed to reach any significant pre–post change (∆-17.26%; ES = −0.20, *p* > 0.05). Regarding the individual responses, our findings indicated that 46.15% of the RNE group (n = 6) and 35.71% of the CG (n = 4) experienced a reduction in their asymmetry scores that exceeded the SWC_0.2_.

## 4. Discussion

This study aimed to assess the effects of an 8-week intervention using the RNE compared with regular training alone on physical fitness measures in male and female youth karate athletes. The key findings demonstrated that a short-term RNE intervention, performed twice weekly, produced significant trivial-to-large improvements in sprint speed, CoD speed, CMJ height, and SLJ distance. Additionally, the results indicated a significant moderate reduction in asymmetry scores within the RNE group. In contrast, no significant changes were observed in the CG across all fitness measures.

Persuasive evidence indicates that eccentric training improves muscle strength, muscle power, and stretch–shortening cycle (SSC) performance more than concentric or traditional (combined eccentric and concentric training) training [5,7,22]. Karate is a dynamic sport where the ability to perform high-velocity movements, particularly kicks and punches, is strongly dependent on the strength and power of the lower limb muscles, especially the knee extensors [1,3]. With that said, eccentric training appears to be an effective method for enhancing key aspects of fitness in karate. The main findings of this study suggest that the RNE, a simple and easy-to-implement eccentric exercise targeting the knee extensors, performed twice per week for 8 weeks, led to significant improvements across all measures of physical fitness. In contrast, regular karate training alone did not appear to provide enough stimuli to elicit significant changes in physical fitness. Specifically, our results showed trivial-to-large increases in 10 m sprint speed, CoD speed, CMJ height, SLJ distance, and asymmetry score (ES = 0.33 to 1.63) in the RNE group. These outcomes are notable, given that the RNE sessions lasted only between 10 and 20 min. This suggests that incorporating a short eccentric exercise like the RNE into regular karate training could significantly enhance various key physical fitness measures relevant to the sport.

An additional key aspect explored in this study was the individual responses to RNE training in both groups [23]. Our findings revealed that, in the RNE group, all karate athletes displayed improvements in linear sprint speed and CoD speed beyond the SWC_0.2_ thresholds. In contrast, within the CG, 53.84% of participants improved their linear sprint speed beyond the SWC_0.2_ and only 7.14% exceeded the SWC_0.2_ for CoD speed. RNE training consistently outperformed regular karate training across the other fitness measures (i.e., CMJ height, SLJ distance, and asymmetry score), with 46.15% to 76.92% of RNE participants achieving improvements above the SWC_0.2_. In the CG, 12.29% improved CMJ height, 35.71% showed enhanced asymmetry scores, and no participants exceeded the SWC_0.2_ in horizontal jump performance. These results suggest that, while regular karate training may lead to some individual improvements above the SWC_0.2_, integrating RNE training produces significant, more consistent gains in all or the majority of participants.

Although speculative, the improvements in power, linear sprint speed, and CoD speed performance likely result from both neural adaptations and morphological changes in the muscle tendon unit following RNE training [4]. There is substantial evidence that eccentric training elicits a greater increase in muscle cross-sectional area compared with concentric or traditional training [4,24]. Additionally, there are indications that eccentric training increases the number of sarcomeres in series [25,26,27] and promotes the selective hypertrophy of fast-twitch muscle fibers [24,28,29]. While these physiological adaptations were not directly measured in this study, they likely contributed to the observed improvements in power, linear sprint speed, and CoD speed performance. This could be supported by the study of Alonso-Fernández and Fernández-Rodríguez [18]. They investigated the effects of the RNE on the architectural adaptations of the rectus femoris in young adults with an average age of 24 years. This study found that an 8-week RNE training program, conducted twice per week, led to substantial increases in muscle fascicle length, muscle thickness, pennation angle, and cross-sectional area [18].

Of note, the findings from this study showed that RNE training led to a significant reduction in the asymmetry score (ES = 0.64). This finding is particularly important, as karatekas are exposed to lower limb injuries [30], and reducing lower limb asymmetry may help mitigate injury risk [31,32]. In contrast, the CG experienced a worsening in the asymmetry score, though this change was not statistically significant (ES = −0.20). To the best of the authors’ knowledge, this is the first study to examine the effects of the RNE on physical fitness and asymmetry in youth karate athletes, leaving no prior data for direct comparison. In soccer, for instance, the asymmetry score is a potential risk factor for injuries and can negatively affect performance [31]. Future studies are needed to investigate the impact of asymmetry on injury occurrence in karate and to determine whether reducing asymmetry scores could help decrease injury rates.

### Limitations

This study has a number of limitations that warrant discussion. First, physical performance was assessed using field tests without any physiological measures. (e.g., musculotendinous morphological/architectural changes) that can provide insights about the mechanisms underpinning the reported adaptations. Additionally, due to time constraints and logistic reasons, it was not possible to monitor the neural changes such as by measuring muscle activation (e.g., electromyography). Second, a quasi-randomization process, rather than a pure randomization method, was employed in this study. Quasi-random methods often involve allocating participants based on specific criteria or non-random factors, which can introduce selection bias. As such, future studies adopting a pure randomization process are needed.

## 5. Conclusions

The main findings of this study show that a short-term RNE intervention, performed twice weekly for 10 to 20 min per session, led to significant trivial-to-large improvements in sprint speed, CoD speed, CMJ height, and SLJ distance. In contrast, no significant changes were observed following regular karate training across these fitness measures. Additionally, the RNE group experienced a significant moderate improvement in asymmetry scores, while regular karate training led to a non-significant worsening. These results suggest that the RNE is an effective and feasible training method that can be regularly integrated into youth karate training routines. Future studies should investigate the combined effects of the RNE and Nordic hamstring exercises on physical fitness and morphological/neural adaptations in youth karatekas.

## Figures and Tables

**Figure 1 jfmk-09-00265-f001:**
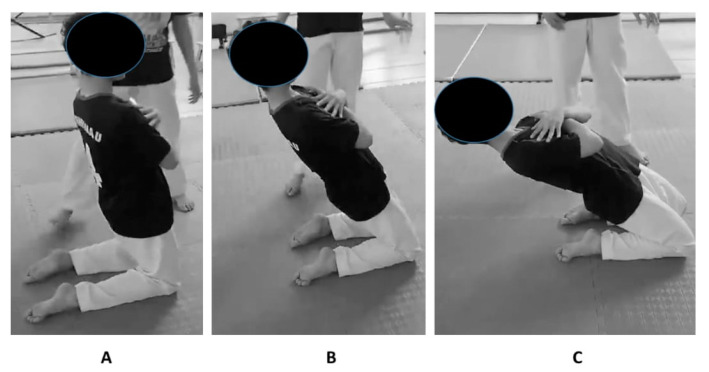
The reverse Nordic exercise with the starting (**A**), mid (**B**) and end position (**C**).

**Table 1 jfmk-09-00265-t001:** Anthropometric characteristics of the included participants.

	RNE Group (n = 13)	CG (n = 14)
Age (years)	15.35 ± 1.66	15.30 ± 1.06
Body height (cm)	1.67 ± 0.07	1.69 ± 0.08
Body mass (kg)	63.85 ± 6.70	65.11 ± 8.88
Maturity offset (years) *	1.41 ± 1.58	2.07 ± 1.27
APHV (years)	13.39 ± 0.81	13.27 ± 1.18

Notes: Data are presented as means and standard deviations; RNE = reverse Nordic exercise group; CG = control group; *: as years from peak height velocity. APHV = age at peak height velocity.

**Table 2 jfmk-09-00265-t002:** Reverse Nordic hamstring exercise program.

Week	Session per Week	Sets	Repetition	Rest Between Sets (s)
1	2	2	6	90
2	2	4	6	90
3	2	4	6	90
4	2	4	8	90
5	2	4	10	90
6	2	4	10	90
7	2	4	10	90
8	2	4	10	90

**Table 3 jfmk-09-00265-t003:** Group-specific changes in measures of physical fitness in both groups before and after eight weeks of eccentric or regular karate training.

	RNE Group (n = 13)				CG (n = 14)				ANOVA	
	Pretest		Posttest		Pretest		Posttest		*p*-Value (ES)	
	M	SD	M	SD	M	SD	M	SD	Time	Group × Time
10 m sprint (s)	2.33	0.17	2.09	0.12	2.22	0.13	2.22	0.18	<0.001 (2.94)	<0.001 (2.89)
505 CoD speed (s)	2.97	0.19	2.75	0.18	3.08	0.19	3.10	0.21	<0.01 (1.59)	<0.001 (1.94)
CMJ (cm)	28.72	6.35	32.44	7.23	27.55	7.12	28.16	6.88	<0.001 (1.96)	<0.01 (1.43)
SLJ (m)	1.90	0.27	1.98	0.27	1.81	0.26	1.78	0.27	<0.01 (1.66)	<0.01 (1.32)
Asymmetry score (%)	4.65	4.10	2.47	2.39	4.98	3.48	5.84	5.41	>0.05 (1.23)	<0.05 (1.03)

M: mean; SD: standard deviation; RNE: reverse Nordic exercise group; CG: control group; ES: effect size; CMJ: countermovement jump; SLJ: standing long jump.

## Data Availability

The original data presented in the study are openly available in FigShare at https://doi.org/10.6084/m9.figshare.27852213.

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
