# Peer review of "The Effects of Reverse Nordic Exercise Training on Measures of Physical Fitness in Youth Karate Athletes"

_jfmk, 2024, doi:10.3390/jfmk9040265_

Round 1

Reviewer 1 Report

Comments and Suggestions for Authors

The authors have prepared a well written manuscript that only faces some minor edits

L56: start with: Previous eccentric training.....conducted primarily with young adults.    

L64: what is CoD?  It hasn't been spelled out first

L72: there is a different font size here

Try to avoid 1st person:

L76: (Instead): To date only one study....

L91 & 96 & 319: avoid "we"

L78 and L299: lowercase rectus femoris

L79: is this mean age for the population?

L113: I think having a mixed sex sample could be viewed as a strength since your outcomes showed significance.  Did you write about that - that maybe the results could be generalized across the sexes?

Could it be seen as a limitation at all?  

When writing ICC - you should use the ICC subscript (e.g.,  3,3)

L184: would including a figure or figures be valuable?  I know of this exercise, but I don't use it a lot

Line 303: delete "More specifically, our"

The findings from this study showed..

L317: delete more

Table 1: what is the * for?

Table 3: create some space between (s) and 2.97

Author Response

Reviewer 1 (Changes highlighted in green)

Comment 1: The authors have prepared a well written manuscript that only faces some minor edits

Response: Thank you for your positive feedback. Highly appreciated.

Comment 2: L56: start with: Previous eccentric training.....conducted primarily with young adults.    

Response: Amended as suggested. The revised statement now reads:

“Previous eccentric training studies have primarily been conducted with young adults [4].”

Comment 3: L64: what is CoD?  It hasn't been spelled out first

Response:  Thank you for pointing this out. CoD refers to change of direction. This is now clarified in the revised version of the manuscript.

Comment 4: L72: There is a different font size here

Response: Sorry for this. It seems that this for some reason happened when saving the document from docx to pdf.

Comment 5: Try to avoid 1st person

Response: We have adopted your suggestion across the entire manuscript. Thank you.

Comment 6: L76: (Instead): To date only one study....

Response: Modified as suggested. Thank you.

Comment 7: L91 & 96 & 319: avoid "we"

Response: Done as suggested.

Comment 8: L78 and L299: lowercase rectus femoris

Response: Corrected. Thank you

Comment 9: L79: is this mean age for the population?

Response: Correct. That’s the mean age of the population.

Comment 10: L113: I think having a mixed sex sample could be viewed as strength since your outcomes showed significance.  Did you write about that - that maybe the results could be generalized across the sexes?

Could it be seen as a limitation at all?  

Response: Thank you for your comment. The point of the reviewer is legitimate. However, the sample size of both the experimental and control groups could be considered adequate for such a study but it is definitely not large enough to make claims about the broader generalizability of the findings. To date, this is the first study that explored the effects of reverse Nordic exercise on measures of physical fitness. As such, although we could make such a claim, we think it is somehow premature. In the end, this study offers good (first) indications related to the effectiveness of the reverse Nordic exercise on various measures of physical fitness in both male and female youth athletes.

Comment 11: When writing ICC - you should use the ICC subscript (e.g., 3,3)

Response: Thank you for your comment. We have clarified the ICC model used, which is ICC(3,1). This is due to the fact that two individual trials were conducted for each test. Upon reviewing the first version of the manuscript, we noticed that three trials were incorrectly indicated for some tests, leading to inconsistency. We apologize for this mistake, which has now been corrected across all tests. We agree with the reviewer that ICC (3,3) would be appropriate if the reliability assessment were based on the average of three trials.

Comment 12: L184: would including a figure or figures be valuable?  I know of this exercise, but I don't use it a lot

Response: In accordance with  the reviewer’s suggestion, we have added a figure illustrating the starting, mid and final positions for the  reverse Nordic exercise.Please refer to the revised version of the manuscript. Thank you.

Comment 13: Line 303: delete "More specifically, our"

The findings from this study showed.

Response: Deleted as suggested. The revised statement reads: “Of note, the findings from this study showed that RNE training led to a significant reduction in the asymmetry score (ES = 0.64)”

Comment 15: L317: delete more

Response: Deleted as suggested

Comment 16: Table 1: what is the * for?

Response: Thank you for bringing this to our attention. The asterisk (*) has now been incorporated into the table for clarity. Its purpose is to provide an explanation of the term 'maturity offset'

Comment 17: Table 3: create some space between (s) and 2.97

Response: I have to admit that the format has been changed when the docx version was saved as pdf. The (s) is meant to be in front of “505-CoD speed” and not the 2.97 value.  

Reviewer 2 Report

Comments and Suggestions for Authors

General comments

·      Please use consistent font and size format throughout the manuscript

·      Please avoid unnecessary abbreviations.

Abstract

·      Line 25: what kind of asymmetry? Please clarify

·      Line 28: around 47% asymmetry?? Please clarify.

Introduction

·      Line 46-48: This is also valid when compared to traditional conc-ecc training as in DOI: 10.1519/JSC.0000000000004039

·      Overall, I suggest reducing the introduction and avoiding unrelated topics like injury prevention, while instead focusing on the characteristics of the reverse Nordic hamstrings and why these are expected to affect the dependent parameters. Please revise the introduction

Methods

·      Why did the authors base the sample size calculation on 1) a study using a different training program and 2) CMJ as primary dependent variable and 3) 0.85 as d effect size (usually this is not the effect size that the software requires)? Additionally, what software was used?

·      It is not clear what the control group did, so to clarify what additional training the experimental group also performed.

·      I am not sure asymmetry could assessed

Discussion

·      Overall, the discussion is not consistent with the aim and again speculates on topics not directly related to the results. Please revise it and make it consistent with the introduction (once revised) and results.

·      Line 316; delete “functional”

Author Response

Reviewer 2 (Changes highlighted in yellow)

Comment 1: Please use consistent font and size format throughout the manuscript

Response: Sorry for this. It seems that this, for some reason, happened when saving the document from docx to pdf.

Comment 2: Please avoid unnecessary abbreviations.

Response: Thank you for your comment. We have reduced some of the abbreviations and ensured that all the existing ones are well introduced at first appearance.

Abstract

Comment 3: Line 25: what kind of asymmetry? Please clarify

Response: Thank you for your comment. We have included more details as follows:

To track the changes in measures of physical fitness before and after training, tests to assess linear sprint speed (i.e., 10-m), change of direction (CoD) speed (i.e., modified 505 CoD), vertical jumping (i.e., countermovement jump [CMJ] height), and horizontal jumping distance (i.e., standing long jump [SLJ]), and lower-limb asymmetry score (i.e., the difference between SLJ-dominant and non-dominant leg) were carried out”

Comment 4: Line 28: around 47% asymmetry?? Please clarify.

Response: Thank you for your comment. The 46.96% reflects the decrease in lower limb asymmetry from pre-to-posttest in the experimental group. We revised the respective sentence as follows:

Additionally, the asymmetry score exhibited a moderate decrease from pre-to-posttest (∆46.96%, ES=0.64).”

Introduction

Comment 5: Line 46-48: This is also valid when compared to traditional conc-ecc training as in DOI: 10.1519/JSC.0000000000004039 Add to Citavi project by DOI

Response: Thank you for pointing this out. We have integrated the suggested reference and the revised statement reads:

It has been shown to produce greater gains in muscle mass and strength compared to other forms of muscle action (e.g., concentric only, isometric only, or concentric-eccentric) [4, 5, 8, 9].”

Comment 6: Overall, I suggest reducing the introduction and avoiding unrelated topics like injury prevention, while instead focusing on the characteristics of the reverse Nordic hamstrings and why these are expected to affect the dependent parameters. Please revise the introduction

Response: Thank you for your comment. Studies investigating the reverse Nordic exercise are indeed limited. To the best of our knowledge, only one study (Alonso-Fernández et al., 2018) has examined the effects of reverse Nordic exercise on the architectural adaptations of the rectus femoris in young adults. Details of this study are provided in the introduction. Furthermore, we have included a rationale in the manuscript, explaining why the reverse Nordic exercise might have beneficial effects on measures of physical fitness. Please refer to the following:

To date, only one study has examined the effects of RNE on the architectural adaptations of the rectus femoris in young adults aged 24 years [18]. The findings indicated that 8 weeks, two sessions per week of RNE training generated a large increase in muscle fascicle length, muscle thickness, pennation angle, and cross-sectional area [18]., The structural adaptations indicated in the study of Alonso-Fernández, Fernández-Rodríguez [18] imply that RNE has the potential to improve general physical fitness attributes, such as lower-limb muscle power and CoD speed, that are beneficial across many athletic disciplines, including karate. However, this is yet to be empirically proved.”   

Regarding the point on injury prevention, the idea was to align the rationale (introduction) with the methods (lower limb asymmetry as a dependent variable) as well as the discussion. Research indicates that reducing lower limb asymmetry may help mitigate injury risk (Chalmers et al., 2017; Bishop et al., 2023). Additionally, there is robust evidence that eccentric training can reduce the risk and severity of injuries (Beato et al., 2021; Peterson et al., 2011; Hu et al., 2023). Consequently, we wanted to explore whether the reverse Nordic exercise could influence lower limb asymmetry, potentially contributing to a reduction in injury risk. We hope our explanation clarifies the point effectively. However, we remain open to further discussion on this aspect with the reviewer.

References

Chalmers S, Fuller JT, Debenedictis TA, Townsley S, Lynagh M, Gleeson C, et al. Asymmetry during preseason Functional Movement Screen testing is associated with injury during a junior Australian football season. J Sci Med Sport. 2017 Jul;20(7):653-7.

Bishop, C, de Keijzer, KL, Turner, AN, and Beato, M. Measuring interlimb asymmetry for strength and power: A brief review of assessment methods, data analysis, current evidence, and practical recommendations. J Strength Cond Res 37(3): 745–750,2023

Beato M, Maroto-Izquierdo S, Turner AN, Bishop C. Implementing Strength Training Strategies for Injury Prevention in Soccer: Scientific Rationale and Methodological Recommendations. International journal of sports physiology and performance. 2021 Mar 1;16(3):456-61.

Petersen J, Thorborg K, Nielsen MB, Budtz-Jørgensen E, Hölmich P. Preventive effect of eccentric training on acute hamstring injuries in men's soccer: a cluster-randomized controlled trial. The American journal of sports medicine. 2011 Nov;39(11):2296-303.

Hu C, Du Z, Tao M, Song Y. Effects of Different Hamstring Eccentric Exercise Programs on Preventing Lower Extremity Injuries: A Systematic Review and Meta-Analysis. International journal of environmental research and public health. 2023 Jan 23;20(3).

Alonso-Fernández D, Fernández-Rodríguez R, Abalo-Núñez R. Changes in rectus femoris architecture induced by the reverse nordic hamstring exercises.e Journal of Sports Medicine and Physical Fitness. 2018.

Methods

Comment 7: Why did the authors base the sample size calculation on 1) a study using a different training program and 2) CMJ as primary dependent variable and 3) 0.85 as d effect size (usually this is not the effect size that the software requires)? Additionally, what software was used?

Response: Thank you for your comment. As previously stated, no studies have examined the effects of reverse Nordic exercise on fitness measures in youth. In addition, although the study used to calculate the sample size may look different in terms of the eccentric exercise applied (i.e., Nordic hamstring exercise), it examined its effects on similar measures of physical fitness within a similar age group. We chose CMJ because it was considered in both studies, and eccentric training showed beneficial effects on this variable. We used the G*Power software and this is now clarified in the manuscript. Coming back to the third point, the G*Power software uses Cohen's f instead of Cohen's d. We have therefore converted Cohen's d (0.85) to Cohen's f (0.42). This means that the value we used for the software was 0.42. We hope our explanation has effectively clarified these points.

Comment 8: It is not clear what the control group did, so to clarify what additional training the experimental group also performed.

Response: Thank you for your comment, and we apologize for not clarifying this important aspect in the manuscript. Both groups participated in four karate training sessions per week. The intervention group replaced 10 to 20 minutes of low-intensity karate drills with the reverse Nordic exercise on Tuesdays and Thursdays, performed immediately after the standard warm-up. The control group followed their usual karate training routine. Therefore, the total training exposure was comparable between the two groups. We added more details in this sense in the revised version as follows:

“The CG followed their usual karate training routine. Therefore, the total training exposure was comparable between the two groups.”

Comment 9: I am not sure asymmetry could assessed

Response: We are afraid that this comment is a bit unclear. If the question is related to the way lower limb asymmetry was calculated, we would refer the reviewer to the following:

To determine the asymmetry score, the SLJ-dominant leg and SLJ-non-dominant leg were conducted while participants adopted a standing position on the designated testing leg, with their hands on hips and their toes behind the starting line. The dominant leg is identified as the preferred leg that the practitioner uses for executing kicks. Participants were then instructed to hop forward as far as possible and land on the same leg. Upon landing, participants were required to ‘hold and stick’ their position for ~ 2 s. Mean inter-limb asymmetry was calculated as a percentage difference between limbs in the unilateral tests using the following equation: (100 / [maximum value] × [minimum value] × -1 + 100) [20]. The between-trial ICCs were 0.92, 0.93 for the SLJ-dominant leg, and SLJ-non-dominant leg, respectively.”

However, if the reviewer intended something different, we kindly request clarification of the comment, and we would be happy to address it accordingly.

Discussion

Comment 10: Overall, the discussion is not consistent with the aim and again speculates on topics not directly related to the results. Please revise it and make it consistent with the introduction (once revised) and results.

Response: Thank you for your comment. The reviewer’s feedback appears somewhat vague and lacks specific details. Based on the previous remarks, we infer that the reference to “topics not directly related to the results” may pertain to the injury prevention aspects mentioned in the introduction. To address this, we clarified this point in our response to comment 6.

The discussion is structured as follows: Summary of Main Findings: The first paragraph offers a concise summary of the study's key results. Relevance of Eccentric Training: The second paragraph contextualizes the significance of eccentric training in karate and discusses the study outcomes. Individual Training Responses: The third paragraph addresses individual variability in training responses across both the experimental and control groups. Potential Mechanisms: The fourth paragraph delves into the mechanisms potentially driving physical fitness adaptations associated with the reverse Nordic exercise. While acknowledging the lack of direct physiological measures as a limitation, we believe discussing potential mechanisms based on previous studies enriches the interpretation of the findings. Injury Risk Mitigation: The final paragraph explores the relationship between reduced lower limb asymmetry and injury prevention, incorporating relevant literature. While these outcomes remain exploratory, we emphasize that further studies are required to establish definitive conclusions regarding the effects of the reverse Nordic exercise on injury incidence and lower limb asymmetry. We hope we managed to clarify the logic behind the chosen structure for the discussion. We remain open to further feedback or suggestions for improvement.

Comment 11: Line 316; delete “functional”

Response: Deleted as recommended and replaced with the word “physical”.

Round 2

Reviewer 2 Report

Comments and Suggestions for Authors

No further comment.